# ATTACKING FEW-SHOT CLASSIFIERS WITH ADVERSARIAL SUPPORT SETS

## ABSTRACT

Few-shot learning systems, especially those based on meta-learning, have recently made significant advances, and are now being considered for real world problems in healthcare, personalization, and science. In this paper, we examine the robustness of such deployed few-shot learning systems when they are fed an imperceptibly perturbed few-shot dataset, showing that the resulting predictions on test inputs can become worse than chance. This is achieved by developing a novel *Adversarial Support Set Attack* which crafts a poisoned set of examples. When even a small subset of malicious data points is inserted into the support set of a meta-learner, accuracy is significantly reduced. For example, the average classification accuracy of CNAPs on the Aircraft dataset in the META-DATASET benchmark drops from $69.2\%$ to $9.1\%$ when only $20\%$ of the support set is poisoned by imperceptible perturbations. We evaluate the new attack on a variety of few-shot classification algorithms including MAML, prototypical networks, and CNAPs, on both small scale (*mini*ImageNet) and large scale (META-DATASET) few-shot classification problems. Interestingly, adversarial support sets produced by attacking a meta-learning based few-shot classifier can also reduce the accuracy of a fine-tuning based classifier when both models use similar feature extractors.

## 1 INTRODUCTION

Standard deep learning approaches suffer from poor sample efficiency (Krizhevsky et al., 2012) which is problematic in tasks where data collection is difficult or expensive. Recently, few-shot learners have been developed which address this shortcoming by supporting rapid adaptation to a new task using only a few labeled examples (Finn et al., 2017; Snell et al., 2017). This success has meant that few-shot learners are becoming increasingly attractive for real-life applications. They have been applied to user personalization in recommender systems (Lee et al., 2019), matching potential users to businesses (Li et al., 2020), personalized talking head models (Zakharov et al., 2019), and on-device gaze estimation (He et al., 2019). As few-shot learners improve, they are also being applied to increasingly sensitive applications where the repercussions of confidently-wrong predictions are severe. Examples include clinical risk assessment (Sheryl Zhang et al., 2019), glaucoma diagnosis (Kim et al., 2017), identification of diseases in skin lesions (Mahajan et al., 2020), and tissue slide annotation in cancer immuno-therapy biomarker research (Lahiani et al., 2018).

As few-shot learners gain popularity, it is essential to understand how robust they are and whether there are potential avenues for their exploitation. It is well known that standard classifiers are vulnerable to inputs that have been purposefully modified in a minor way to cause incorrect predictions (Biggio & Roli, 2017). Such examples may be presented to a model either at test time, called *evasion attacks* (Biggio et al., 2017) or *adversarial examples* (Szegedy et al., 2014), or at training time, which is referred to as *poisoning* (Newsome et al., 2006; Rubinstein et al., 2009). While previous work has considered adversarial attacks on few shot learners, data poisoning attacks have not been studied and are the focus of this paper.

Data poisoning attacks are of particular relevance in the few-shot learning setting for two reasons. First, since the datasets are small, a handful of poisoned patterns might have a significant effect. Second, many applications of few-shot learning require labeled data from users to adapt the system to a new task, essentially providing a direct interface for outsiders to influence the model's behaviour.

If few-shot learning systems are not robust to poisoning of their training dataset, then this weakness could be exploited. An attacker performing a man-in-the-middle (Conti et al., 2016) data poisoning attack could cause a recommender system's personalization to perform badly or suggest certain results to influence a user's decision. Applied at scale to many users, an attacker could cause significant damage. Similarly, a doctor attempting to commit medical insurance fraud may submit images causing a benign skin condition to be incorrectly classified as a skin disease requiring expensive treatment; or a malicious party may ruin a research study that uses automated annotation of samples by tampering imperceptibly with only a few images. If these attacks could be achieved with malicious patterns that cannot be reliably distinguished from real training data, it would be difficult to defend against them.

Before detailing the key contributions of the paper, it is necessary to briefly introduce the lexicon of few-shot learning. During training, few-shot learners are typically presented with many different tasks. The model must learn to perform well on each task, hopefully arriving at a point where it can adapt effectively to a new task at test time. At test time, the model is presented with an unseen task containing a few labeled examples, the *support set*, and a number of unlabeled examples to classify, called the *query set*. The paper makes the following contributions:

1. We define a novel attack on few-shot classifiers, called an *Adversarial Support Set Attack*, which applies adversarial perturbations to the support set that are calculated to minimize model accuracy over a set of query points. To the best of the authors' knowledge, this is the first work considering the impact of poisoning attacks on trained few-shot classifiers.

2. We demonstrate that few-shot classifiers are surprisingly vulnerable to Adversarial Support Set attacks. The adversarial support set attack is more effective than the baselines considered, and generalizes well, i.e. the compromised classifier is highly likely to be inaccurate on a randomly sampled query set from the task domain.

3. We demonstrate the effectiveness of our approach against a variety of few-shot classifiers including MAML (Finn et al., 2017), ProtoNets (Snell et al., 2017), and CNAPs (Requeima et al., 2019a), on both small scale (*mini*ImageNet (Vinyals et al., 2016)) and large scale (META-DATASET (Triantafillou et al., 2020)) few-shot classification benchmarks.

4. We show that adversarial support sets transfer effectively to fine-tuning based few-shot classifiers when the few-shot classifier and the fine-tuner utilize similar feature extractors.

The rest of the paper proceeds as follows: Section 2 provides background about the meta-learning models under consideration, relevant adversarial attack methods, and the threat model under consideration. Section 3 discusses how evasion and poisoning attacks may be generalized to few-shot learners. Section 4 presents the experimental results and Section 5 concludes the paper. Additional results and experimental details are in the Appendix.

## 2 BACKGROUND

In this section we lay the necessary groundwork for adversarial support set attacks. We focus on image classification. We denote input images $x \in \mathbb{R}^{ch \times W \times H}$ where $W$ is the image width, $H$ the image height, $ch$ the number of image channels and image labels $y \in \{1, \ldots, C\}$ where $C$ is the number of image classes. We use bold $\boldsymbol{x}$ and $\boldsymbol{y}$ to denote a set of images and labels, respectively.

### 2.1 META-LEARNING

We consider the few-shot image classification scenario using a meta-learning approach. Rather than a single, large dataset $D$, we assume access to a dataset $\mathcal{D} = \{\tau_t\}_{t=1}^{K}$ comprising a large number of training *tasks* $\tau_t$, drawn i.i.d. from a distribution $p(\tau)$. The data for a task consists of a *support set* $D_S = \{(x_n, y_n)\}_{n=1}^{N}$ comprising $N$ elements, with the inputs $x_n$ and labels $y_n$ observed, and a *query set* $D_Q = \{(x_m^*, y_m^*)\}_{m=1}^{M}$ with $M$ elements for which we wish to make predictions. We may use the shorthand $D_S = \{\boldsymbol{x}, \boldsymbol{y}\}$ and $D_Q = \{\boldsymbol{x}^*, \boldsymbol{y}^*\}$ for brevity. Here the inputs $x^*$ are observed and the labels $y^*$ are only observed during meta-training (i.e. training of the meta-learning algorithm). Note that the query set examples are drawn from the same set of labels as the examples in the support set. At meta-test time, the classifier $f$ is required to make predictions for query set

| Support Set $D_S$ | $N = 4$ | 1 | 2 | 3 | 4 |
|---|---|---|---|---|---|
| | $x$ | | | | |
| | $y$ | stopwatch | clock | watch | meter |

| Query Set $D_Q$ | $M = 4$ | 1 | 2 | 3 | 4 |
|---|---|---|---|---|---|
| | $x^*$ | | | | |
| | $y^*$ | meter | watch | stopwatch | clock |

Figure 1: Example task with $C = 4$ classes with $N = 4$ and $M = 4$.

inputs of unseen tasks. Often, the meta-test tasks will include classes that have not been seen during meta-training, and $D_S$ will contain only a few observations. An example task is shown in Fig. 1.

**Episodic Training**   The majority of modern meta-learning methods employ *episodic* training (Vinyals et al., 2016). During meta-training, a task $\tau$ is drawn from $p(\tau)$ and randomly split into a support set $D_S$ and query set $D_Q$. The meta-learner $g$ takes as input the support set $D_S$ and produces task-specific classifier parameters $\psi = g(D_S)$ which are used to adapt the classifier $f$ to the current task. The classifier can now make task-specific predictions $f(x^*, \psi = g(D_S))$ for any test input $x^* \in D_Q$. Refer to the diagram labeled *Clean* in Fig. 2. A loss function $\mathcal{L}(f(x^*, \psi), y^*)$ then computes the loss between the predictions for the label $f(x^*, \psi)$ and the true label $y^*$. Assuming that $\mathcal{L}$, $f$, and $g$ are differentiable, the meta-learning algorithm can then be trained with stochastic gradient descent by back-propagating the loss and updating the parameters of $f$ and $g$.

**Common Few-shot Learning Algorithms**   There has been an explosion of meta-learning based few-shot learning algorithms proposed in recent years. For an in-depth review see Hospedales et al. (2020). Here, we briefly describe methods that are relevant to our experiments. Arguably the most widely used is the gradient-based approach, the canonical example for modern systems being MAML (Finn et al., 2017). With MAML, the task-specific parameters $\psi$ are the parameters of the classifier $f$ after applying one or more gradient steps taken on the support set $D_S$. Another widely used class of meta-learners are *amortized-inference* or *black box* based approaches e.g, VERSA (Gordon et al., 2019) and CNAPs (Requeima et al., 2019a). In these methods, the task-specific parameters $\psi$ are generated by one or more *hyper-networks*, $g$ (Ha et al., 2016). An important special case of this approach is Prototypical Networks (ProtoNets) (Snell et al., 2017) which is based on *metric* learning and employs a nearest neighbor classifier, and the task-specific parameters $\psi$ are the mean feature vectors for each class $c \in C$ of the support set $D_S$. Finally, a very competitive, non-episodic approach to few-shot learning is simple *fine-tuning* (Yosinski et al., 2014; Tian et al., 2020). In this approach, a pretrained feature extractor is used in combination with a final layer linear classifier, whose weights serve as the task-specific parameters $\psi$ that are learned by taking a number of gradient steps on the support set data.

## 2.2   ADVERSARIAL ATTACKS

Szegedy et al. (2014) have shown that it is possible to craft *adversarial examples*, inputs that are indistinguishable from legitimate examples to both humans and algorithms (Carlini & Wagner, 2017), yet are classified incorrectly by deep neural networks with high confidence (Nguyen et al., 2014). Adversarial examples are typically generated by taking a *clean*, unperturbed input image and adding a small amount of carefully crafted, almost imperceptible noise (Goodfellow et al., 2014; Madry et al., 2017). The perturbation $\delta$ is calculated by performing the following optimization $\arg \max_\delta \mathcal{L}(\theta, x + \delta, y)$, where $\mathcal{L}$ is the loss function e.g. the cross entropy, $\theta$ denotes the model parameters, $x$ is the input being perturbed, and $y$ is its label. The perturbation size is typically constrained by some norm such as $\ell_\infty$ or $\ell_2$. As stated above, the optimization produces an *untargeted* attack, in the sense that we do not require the generated image to be misclassified as any particular class, only to be classified incorrectly. An alternative formulation of the optimization problem, $\arg \min_\delta \mathcal{L}(\theta, x + \delta, \tilde{y})$, would produce a *targeted* attack, i.e. an image that is misclassified as belonging specifically to the class with label $\tilde{y} \neq y$. In this paper, labels for targeted attacks are generated by shifting the clean class label by $1 \mod C$. Many algorithms can be used to generate adversarial examples, including Projected Gradient Descent (Madry et al., 2017), the Carlini and Wagner $\ell_2$ attack (Carlini & Wagner, 2017), or ElasticNet (Chen et al., 2017a). Throughout this paper, without loss of generality, we utilize Projected Gradient Descent (PGD) with an $\ell_\infty$ norm, because is it simple to implement, inexpensive to compute, and effective. We show that by careful application of PGD, we can successfully poison meta-learners at meta-test time.

## 2.3 THREAT MODEL

The threat model may be summarized in terms of the adversary's goal, knowledge and capabilities.

**Goal** The adversary could aim to compromise a system's integrity, availability or confidentiality. Poisoning attacks may aim to compromise system integrity — for example, backdoor poisoning (Chen et al., 2017b), where the aim is to misclassify only a specific query point at test time. In this paper, we consider poisoning attacks that reduce system availability, where the goal is simply to maximize system failure on any query point. Evasion attacks usually aim to compromise the integrity of the system by causing failure on specific query points at test time.

**Knowledge** We assume the adversary has full knowledge of the model's internal workings — including gradients and other internal state information. We also assume that they can access enough data to be able to form a seed query set of at least the same size as the adversarial support set.

**Capabilities** When performing adversarial support set attacks, we consider an adversary who is able to manipulate some fraction of the support set. We constrain the attack-space by requiring that the adversary's modifications must be imperceptible. We achieve this by constraining the adversarial perturbation to be within some $\epsilon$ of the original image, measured using the $\ell_\infty$ norm. As a baseline, we also considered an attacker who is able to modify support set labels instead. See *Other Possible Attacks* in Section 3 for more details on this attack. When performing query attacks, we consider an adversary who is able to manipulate one or more points in the query set, again constrained by a maximum perturbation size.

## 3 ADVERSARIAL ATTACKS ON FEW-SHOT LEARNERS

In this section, we introduce a variety of attacks that can be perpetrated against few-shot learners, summarized in Fig. 2. Consider a few-shot learning system that has been trained and deployed as a service. In this service, users or groups of users provide their data as input to the service, either explicitly such as tagged photos or implicitly such as product preferences or browsing habits. This data forms a support set and is used by the system to produce task-specific parameters which are then used to make predictions on novel queries from the users. A malicious user, or a man-in-the-middle attacker could attack the system in a number of ways, discussed below.

**Query Attack** The attacker may want the adapted classifier to misclassify a specific input image. This corresponds to solving $\arg\max_\delta \mathcal{L}(f(x^* + \delta, g(\boldsymbol{x}, \boldsymbol{y})), y^*)$. Refer to Appendix A.2 for details. These kinds of attacks relate to adversarial examples as considered in recent literature, in the context of evasion attacks (Biggio et al., 2017). Query attacks have been perpetrated successfully against few-shot learners (Goldblum et al., 2019; Yin et al., 2018), but are not the main focus of this work.

**Support Attack** The attacker may want the system to fail on *any* query image. The attacker will achieve this by computing a perturbed support set $\tilde{D}_S = \{\tilde{\boldsymbol{x}}, \boldsymbol{y}\}$ whose inputs are optimized to fool the system on a specific query set, which we call the *seed query set*, with the goal of generalizing to unseen query sets. This corresponds to solving $\arg\max_\delta \mathcal{L}(f(\boldsymbol{x}^*, g(\boldsymbol{x} + \boldsymbol{\delta}, \boldsymbol{y})), \boldsymbol{y}^*)$ such that $\|\boldsymbol{\delta}\|_\infty < \epsilon$, where $D_Q = \{\boldsymbol{x}^*, \boldsymbol{y}^*\}$ denotes the seed query set and $\epsilon$ is the maximum size of the perturbation. Refer to Algorithm 1 for details. We call this novel few-shot learner attack an *Adversarial Support Set Attack*, or simply a *support attack*. Our attack is a poisoning attack, since the attacker is manipulating data that the model will use to do inference. However, it is important to note the the attack is perpetrated at meta-test time, after the meta-learner has already been meta-trained. Example images from a support attack are shown in Fig. 3.

In contrast to a query attack, which only considers the model's behaviour on a single point at a time, support attacks allow the adversarial optimization function to incorporate information regarding the model's behaviour on the entire query set. In real settings, an attacker might design an attack on their own query set, hoping it will generalize to other, unseen queries. The ability to generalize will depend on $M$, the size of the seed query set.

**Adversarial support sets vs evasion attacks** The adversarial support set attack differs from an evasion attack in the following ways: (i) We generate an entire adversarial support set (or subset of

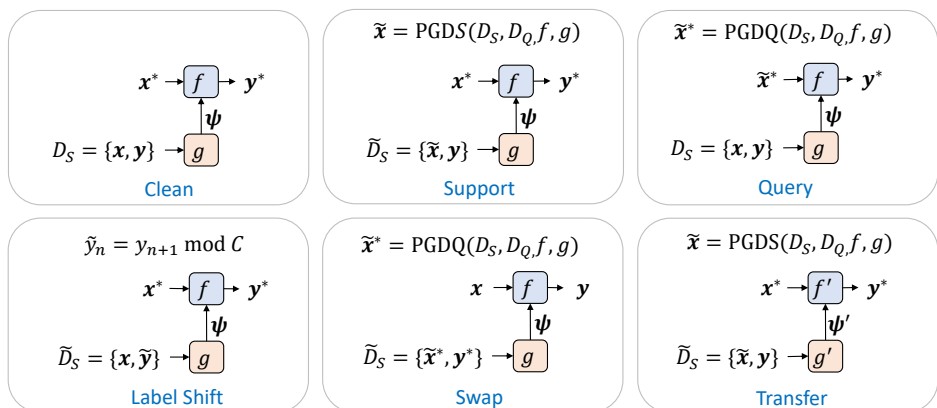

Figure 2: Range of considered attacks on meta-learning based few-shot image classifiers. $f$ and $g$ denote the classifier and trained meta-learner, respectively. Each diagram depicts how an attack is applied and includes an expression for the attack's computation. Adversarially perturbed quantities are denoted with a tilde. The prime ($'$) marks in the Transfer attack indicate that the attack is transferred to a different meta-learner and classifier than the attack is derived from.

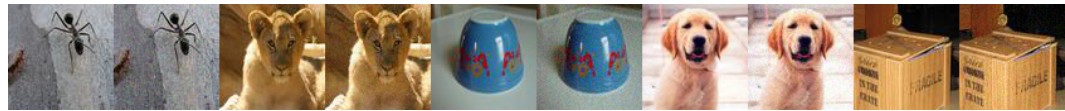

Figure 3: Pairs of images from the *mini*ImageNet dataset where the left is unperturbed, while the right is adversarially perturbed by a PDG *Support* attack with $\epsilon = 0.05$, $\gamma = 0.0015$, and $L = 100$.

the support set) at once, rather than generating just a single adversarial input. (ii) The attack is constructed with regard to an entire query dataset, rather than a single query point. Moreover, the goal is not for a single specific query to be misclassified, but for all future unseen queries to be misclassified. (iii) Generation of the attack involves backpropagating the gradients through a meta-learner which itself is performing gradient based learning in the case of MAML, or is a complex set neural network in the case of ProtoNets or CNAPs. In this context, it is prudent to use a fairly simple attack method in the face of significant increased complexity over the standard evasion attack setting.

**Other Possible Attacks** We formulate a number of other few-shot learner attacks, also depicted in Fig. 2. *Label Shift* is a simple attack on the support set which involves mislabelling the support set images by shifting the true label index by one in a modulo arithmetic fashion. We consider systematic mislabeling in this way to be a strong attack for comparison, though this "attack" may be easily detected in practice. We also consider *Swap* attacks, a support set poisoning attack where a set of images are perturbed with a query attack and then inserted into the support set. Query attacks are typically cheaper to compute, since they do not require back-propagation through the meta-learner. Swap attacks are thus a strong baseline. See Section 4 for further details of the swap attack's implementation. Lastly, we also formalize transfer attacks in the context of few-shot learning, as shown in Fig. 2, *Transfer*, where the adversarial support set is computed on a first few-shot classifier system and is then applied to a different few-shot classifier. An attacker wishing to target a specific system may perform a transfer attack from a surrogate system to the target if the target's internal state or inner workings are unknown, or if attacking the surrogate is computationally cheaper than an attack against the target directly. In particular, our experiments transfer attacks from meta-learning based few-shot learners to an expensive fine-tuning algorithm.

**Related Work** While there has been previous work on adversarial attacks against few-shot learning systems (Goldblum et al., 2019; Yin et al., 2018), attacks that poison the support set have received little attention. Goldblum et al. (2019) devise a technique called adversarial querying which significantly improves robustness against query attacks (refer to Fig. 2). They do not evaluate robustness against support set attacks at meta-test time, though they do test attacking the support set

---

**Algorithm 1** PGD for Adversarial Support Set Attack

---

**Require:**
    $I_{min}$: Minimum image intensity
    $I_{max}$: Maximum image intensity
    $L$: Number of iterations
    $\epsilon$: Perturbation amount
    $\gamma$: Step size
    $D_S \equiv \{\boldsymbol{x}, \boldsymbol{y}\}$
    $D_Q \equiv \{\boldsymbol{x}^*, \boldsymbol{y}^*\}$
    ▷ We use cross-entropy loss for $\mathcal{L}$.

1: **procedure** PGDS($D_S, D_Q, f, g$)
2:     $\boldsymbol{\delta} \sim U(-\epsilon, \epsilon)$
3:     $\tilde{\boldsymbol{x}} \leftarrow \text{clip}(\boldsymbol{x} + \boldsymbol{\delta}, I_{min}, I_{max})$
4:     **for** $i \in 1, ..., L$ **do**
5:         $\boldsymbol{\delta} \leftarrow \text{sgn}(\nabla_{\tilde{\boldsymbol{x}}}\mathcal{L}(f(\boldsymbol{x}^*, g(\tilde{\boldsymbol{x}}, \boldsymbol{y})), \boldsymbol{y}^*))$
6:         $\tilde{\boldsymbol{x}} \leftarrow \text{clip}(\tilde{\boldsymbol{x}} + \gamma\boldsymbol{\delta}, I_{min}, I_{max})$
7:         $\tilde{\boldsymbol{x}} \leftarrow \boldsymbol{x} + \text{clip}(\tilde{\boldsymbol{x}} - \boldsymbol{x}, -\epsilon, \epsilon)$
8:     **end for**
9:     **return** $\tilde{\boldsymbol{x}}$
10: **end procedure**

---

along with the query set during meta-training. Yin et al. (2018) also describe a meta-training regime to increase robustness against query attacks, but only consider the relatively weak Fast Gradient Sign Method attack (Goodfellow et al., 2014). Edmunds et al. (2017) explore the transferability of query attacks between tasks on meta-trained models and find that the attacks are indeed highly transferable, though their experiments were restricted to the MAML algorithm and the relatively simple Omniglot dataset. In this work, we focus on the effects of support set attacks on the performance of few-shot image classifiers using a variety of learning algorithms and challenging datasets.

## 4 EXPERIMENTS

In this section we present our experiments that endeavor to answer the following questions: (i) How vulnerable are few-shot classifiers to adversarial support set attacks? (ii) What are the most effective settings for the support set attack parameters? (iii) Do adversarial support set attacks crafted on a meta-learning based few-shot classifier transfer to a fine-tuning based few-shot classifier? All of our experiments are carried out on meta-trained few-shot classifiers. Refer to Appendix A.1 for details on the meta-training protocols. We divide the experiments into *small-scale* on the *mini*ImageNet dataset (Vinyals et al., 2016) using the MAML and ProtoNets algorithms, and *large-scale* on META-DATASET (Triantafillou et al., 2020) using the CNAPs algorithm.[1]

When perpetrating support attacks, we consider two variations of the loss function: *all*, in which the entire query set is used; and *single*, in which a single, random point in the query set is chosen for the loss calculation at each attack iteration, with the intention that the additional stochasticity may prevent the attack from getting stuck in local optima. We found that the *all* strategy performed best when combined with a targeted attack, whereas the *single* strategy worked better in combination with an untargeted attack. Unless otherwise specified, we use the targeted, *all* loss strategy.

For comparison purposes, we also consider uniform random noise. Uniform noise is added to an image $x$ as follows: $\tilde{x} = x + u$, where $u \sim U(-\epsilon, \epsilon)$ and $U$ denotes the uniform distribution.

### 4.1 SMALL SCALE EXPERIMENTS

**Attack Comparison** Fig. 4 depicts the relative decrease in 5-way classification accuracy due to a variety of attacks for two values of $\epsilon$. We compute the percentage relative decrease in classification accuracy as follows: $100\% \times (a_{clean} - a_{attack})/a_{clean}$ where $a_{clean}$ is the *clean* classification accuracy, and $a_{attack}$ is the classification accuracy after the attack. We consider the model's performance over 500 randomly generated tasks. Each task is constructed by choosing the required number of classes from the dataset labels at random, then choosing the required number of images randomly within each class. Each task is composed of a support set, a seed query set and 50 unseen query sets used for evaluation of the attack. Although the shot of the seed query set may be varied, the query sets used for evaluation are the same size as the support set. The unseen query sets are all guaranteed to be disjoint from the seed query set to avoid information leakage. We proceed as follows: for each task, we generate an adversarial support set using the corresponding seed query set. We then evaluate the adversarial support set on the task's 50 unseen query sets. Note that we make no further

---

[1]Source code for all experiments will be made available upon publication.

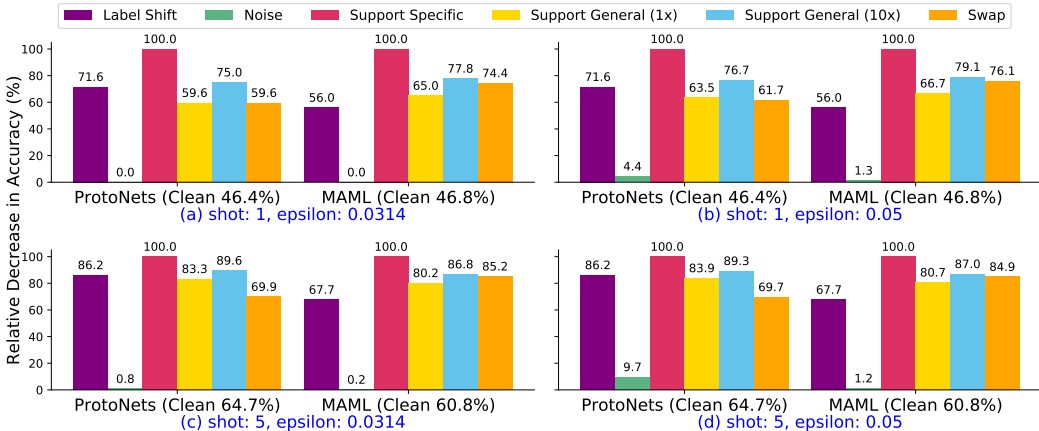

Figure 4: The relative drop in classification accuracy for a variety of attacks against MAML and ProtoNets models in the 5-way *mini*ImageNet configuration, averaged over 500 tasks. For *Support General (1x)*, $M=N$, and for *Support General (10x)*, $M=10N$. All support images were perturbed. PGD settings were $L=100$, with $\gamma = 0.0015$ for $\epsilon = 0.05$, and $\gamma = 9.4\mathrm{e}{-4}$ for $\epsilon = 0.0314$.

changes to the adversarial support set when evaluating it on the unseen query sets. These unseen query sets are also used to evaluate the other attacks and baselines included in Fig. 4. We refer to the average classification accuracy on the seed query sets as the *Support Specific* attack accuracy, whereas the average classification accuracy when evaluating the attack on a random sample of 50 unseen query sets is called the *Support General* attack accuracy. We include two different results for the *Support General* scenario: *Support General (1x)*, for which the seed query set size is the same size as the support set (i.e. $M = N$) and *Support General (10x)*, where $M = 10N$.

When perpetrating a swap attack on a given task, we use the task's support set and query set as in Fig. 2 *Query* to generate a query attack. We then "swap" the role of the sets so that the adversarial query set is presented to the model to learn from i.e. as a support set. We evaluate the swap attack on 50 independent query sets, in the same way as the *Support General* scenario.

The *Support Specific* attack is $100\%$ effective in all cases, indicating that the adversarial objective function is acting as desired. The attack's generalization to unseen query sets, as measured by the *Support General (x10)* attack, successfully beats all the baselines. The *Support General (x1)* attack, is noticeably weaker, but still beats the *Swap* baseline on Protonets. Since we are measuring generalization in these scenarios, it is not unexpected that the attack with less data to learn from does not generalize as well. We further examine the effect of seed query set size later in this section. The *Label Shift* attack causes a large drop in accuracy, but is easily detected by inspection, and so does not provide like-for-like comparison. We consider the *Swap* attack, which is effective and simpler to compute than the adversarial support set attack, to be the strongest competitor. From Fig. 4, there does not appear to be a difference between ProtoNets and MAML in terms of robustness to the adversarial support set attack, though MAML is significantly more vulnerable to the swap attack. For 5-shot classification, increasing the perturbation size does not consistently increase the attack's impact. This is because the step size $\gamma$ was tuned for the 1-shot case, but was not optimal for the 5-shot problem. For consistency, we used the same PGD settings for both in Fig. 4 (details in A.3).

Since we are primarily interested in degrading model accuracy for unseen inputs, we consider only the *Support General* case for the remainder of the results presented, unless otherwise specified.

**Fraction of Poisoned Patterns**  A realistic scenario may constrain an attacker to perturb only a fraction of the images in the support set. We consider two dimensions: specifying the fraction of classes that are adversarially perturbed and, within those classes, how many of the shots are poisoned. Fig. 5a *Support* shows the relative drop in 5-way classification accuracy for ProtoNets as a function of the number of poisoned classes and shots in a support set. As expected, the attack strength increases as the number of poisoned shots are increased, though perturbing even one point causes a significant drop in accuracy. We also see that the attack is stronger when the poisoned points

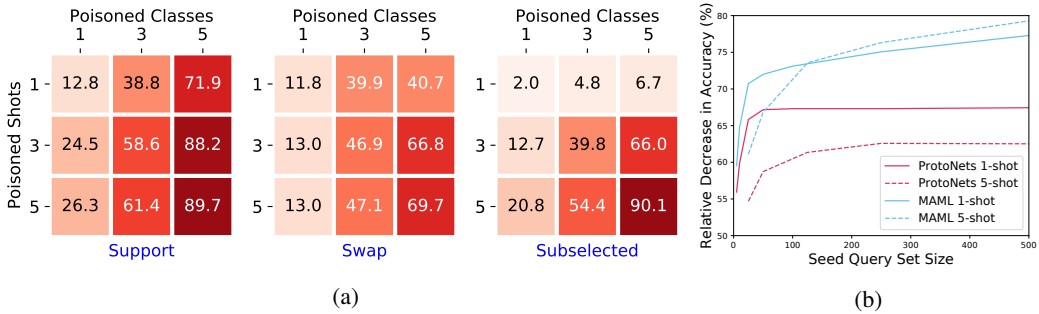

(a)                                                                                          (b)

Figure 5: (a) The relative drop in 5-way, 5-shot classification accuracy of ProtoNets as the number of poisoned classes and poisoned shots within those classes are varied using three different methods of crafting the poisoned images. Darker colors indicate a stronger attack. Attacks were calculated with $\epsilon = 0.05$, $\gamma = 0.0015$, $L = 200$, $M = 13N$, averaged over 250 tasks. (b) Relative drop in 5-way classification accuracy for ProtoNets and MAML using a support attack as $M$ is varied, for 1-shot and 5-shot. Both the targeted, *all* and untargeted, *single* loss strategies were considered and the more effective was chosen for each model. The attacks are generated using PGD with $\epsilon = 0.05$, $\gamma = 0.0015$, $L = 100$, and half the classes and shots in the support set adversarially perturbed.

are spread across classes, e.g. perturbing just one image from every class causes $72\%$ accuracy drop, whereas perturbing all five shots of one class only causes $26\%$ accuracy drop.

For comparison, Fig. 5a *Swap* shows the effect of performing a swap attack with varying fractions of poisoned classes and shots. Poisoning the support set with a single malicious point, either an adversarial support pattern or an adversarial query pattern achieves similar drops in model accuracy. However, increasing the number of poisoned shots for a particular class is significantly more effective for the support set attack than the swap attack because the adversarial support set's joint optimization allows for collusion among its images. We confirm this with an additional experiment shown in Fig. 5a *Subselected*, where we generate an adversarial support set with all patterns poisoned, then select subsets of these perturbed patterns to poison the original support set. Even though the *Support* and *Subselected* attacks have the same fraction of poisoned patterns, the support set attack performs significantly better as it is jointly optimized for a specific fraction to be poisoned, whereas the patterns in the *Subselected* attack have been separated from their colluding images.

**Seed Query Set Size**   Fig. 5b shows that in a low-shot scenario, an attacker can obtain a significantly stronger attack by increasing the seed query set's size, i.e. by increasing the number of examples that the adversarial support set is optimized to fool. The effectiveness of the attack using the ProtoNets algorithm plateaus at $M = 100$ for 1-shot and around $M = 250$ for 5-shot, whereas the effectiveness of a MAML algorithm attack increases steadily with seed query set size. Fig. 5b also shows that ProtoNets is more robust than MAML. ProtoNets is generally a more capable classifier than MAML (Triantafillou et al., 2020). Additionally, MAML uses gradient steps to adapt to new tasks which can be noise-sensitive, whereas ProtoNets computes the mean of the feature vectors for each class in the support set and this averaging tends to reduce noise.

## 4.2   LARGE SCALE EXPERIMENTS

We now attack the CNAPS algorithm when performing classification on a subset of the META-DATASET constituent datasets (refer to Appendix A.1.1 for details on our use of META-DATASET). When following the META-DATASET protocol, task support sets may be large (up to 500 images across all classes). We thus generate a support set attack with approximately $20\%$ poisoned shots per class, which is generally considered the upper limit of manipulated patterns in conventional poisoning approaches (Jagielski et al., 2018), which use significantly larger datasets than few-shot learning. In Fig. 6, we show the relative decrease in accuracy of CNAPS when under attack (unnormalized results are in A.4). *Support Specific* reduces the model's relative accuracy to almost zero for all datasets except MNIST. MNIST is the easiest classification problem in the large-scale suite because there are only ten classes and the input images are simplistic. The *Support General* attack

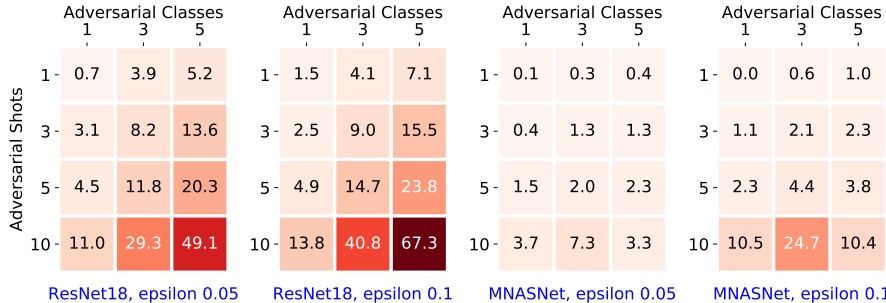

Figure 6: The relative drop in model accuracy for a variety of attacks using the CNAPs algorithm on META-DATASET with $\epsilon = 0.05$, $\gamma = 0.0015$, $L = 100$, averaged over 500 tasks, with all classes, but only 20% of the shots poisoned.

| | Adversarial Classes 1 3 5 | Adversarial Classes 1 3 5 | Adversarial Classes 1 3 5 | Adversarial Classes 1 3 5 |
|---|---|---|---|---|
| 1 | 0.7  3.9  5.2 | 1.5  4.1  7.1 | 0.1  0.3  0.4 | 0.0  0.6  1.0 |
| 3 | 3.1  8.2  13.6 | 2.5  9.0  15.5 | 0.4  1.3  1.3 | 1.1  2.1  2.3 |
| 5 | 4.5  11.8  20.3 | 4.9  14.7  23.8 | 1.5  2.0  2.3 | 2.3  4.4  3.8 |
| 10 | 11.0  29.3  49.1 | 13.8  40.8  67.3 | 3.7  7.3  3.3 | 10.5  24.7  10.4 |
| | ResNet18, epsilon 0.05 | ResNet18, epsilon 0.1 | MNASNet, epsilon 0.05 | MNASNet, epsilon 0.1 |

Figure 7: Relative drop in 5-way classification accuracy when transferring adversarial support attacks as a function of poisoned shots and classes using CNAPs, which uses a ResNet18 network, to fine-tuners that use ResNet18 and MNASNet networks on ILSVRC 2012 from META-DATASET. The attacks were generated with $\gamma = 0.0015$, $L = 100$, $M = 6N$, and averaged over 100 tasks.

significantly impacts classification accuracy, easily out-performing all the baselines, in spite of the fact that only 20% of the support set shots are poisoned. Our results demonstrate that an attacker could cripple a few-shot learning system, even in this more realistic scenario. The large impact of the attack may be partially due to the larger context sets, which allows for more poisoned patterns in total than the small-scale experiments; and partially because these classification benchmarks are much harder problems, with more classes, resulting in decision boundaries that are more fragile.

**Attack Transfer** Fig. 7 demonstrates that poisoning attacks developed for meta-learners can also impede the learning of fine-tuning based few-shot classifiers. Details of the fine-tuning protocol which involves learning the weights of the final layer classifier and FiLM layers (Perez et al., 2018) that modulate pretrained feature extractor weights are in Appendix A.1.5. Fig. 7 shows the effect of transferring the adversarial support sets from the meta-learning based CNAPs algorithm (with a ResNet18 backbone) to fine-tuning based few-shot classifiers (with Resnet18 or MNASNET backbones), as the number of poisoned classes and shots are varied. The MNASNet fine-tuner is largely robust to the adversarial support set, potentially because the feature extractor is very different to that in CNAPs, though attacks with large perturbations and a large percentage of poisoned patterns do have some impact. Transferring the attack from CNAPs to the ResNet18 fine-tuner is more successful, with effects increasing significantly as the percentage of poisoned patterns are increased.

## 5 CONCLUSIONS

This paper introduced the Adversarial Support Set Attack, that attacks trained few-shot classifiers at meta-test time. We show that this attack is more effective than a number of baselines using a variety of few-shot learning approaches, causing predictions on test inputs to be worse than chance. Future work may consider how to defend few-shot classifiers against adversarial support sets, for example by developing analogues of adversarial training algorithms, and investigating the effects of transferring adversarial support sets to conventional deep classifiers at training time.

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
