# OpenReview forum: "Attacking Few-Shot Classifiers with Adversarial Support Sets"
_ICLR.cc/2021/Conference — Reject_

### Official Review · AnonReviewer2 · 2020-10-26
**Interesting ideas, questionable contributions**

**Rating:** 6
**Confidence:** 3

**Review:**

This work proposes adversarial attacks on few-shot learning systems. Evasion attacks are developed which can be viewed a simple applications of PGD/FGSM. The authors then apply evasion attacks to a support dataset, such that the n-shot classifier's loss is maximised on a query set. Additionally, poisoning attacks that cause the classifier to learn a shifted label set are introduced. Experiments on miniImageNet are performed with two meta-learning algorithms (MAML and ProtoNets) showing that these attacks lead to a large decrease in accuracy in comparison to baselines such as random noise attacks or universal adversarial perturbations (UAP) [1]. The authors then perform additional experiments such as measuring drops in accuracy for different numbers of n in n-shot learning and different query set sizes.


I am not meta-learning expert, so I have assumed the authors choice of meta-learning algorithms and datasets is fair. Overall, I thought the paper was well-written, and the experiments were sound and relatively convincing. The downside is that this work introduces attacks, which as far as I can tell, are quite simple extensions or applications of gradient based evasion attacks such as PGD. I'm not sure if this represents a large enough contribution to warrant acceptance. The main contribution seems to lie in experiments showing how little data is required to make these kinds of attacks work, but what should we infer from e.g. figure 4a beyond "if an adversary controls more data the attack is stronger"? Is this a fair characterisation of the paper's contributions? I also have a few areas of confusion I hope the authors can clarify:


1.  I found it curious that the UAP performed so poorly in comparison to the support attack, since the formulations are quite similar. I didn't understand how the UAP was constructed, it seems from Section 4.1 the perturbation was constructed from a separate classification task on 712 training labels from the full ImageNet dataset. Why was the UAP not constructed on the miniImageNet dataset? Or at least on a smaller task on the full ImageNet dataset? It is hard to reason about the veracity of comparison between the two attacks currently.

2. Why was there no comparison with related work? For example, to (Goldblum et al., 2019; Yin et al., 2018) on the evasion attack side?

3. Why is ProtoNets more robust than MAML?


[1] Moosavi-Dezfooli, Seyed-Mohsen, et al. "Universal adversarial perturbations." Proceedings of the IEEE conference on computer vision and pattern recognition. 2017.

---

> ### Author Response · Authors · 2020-11-18
> **Response to Reviewer #2 (Part 1)**
>
> We thank the reviewer for their time in reading and reviewing our paper. Prompted by the reviewer’s questions, we have added further discussions on a number of points, specifically regarding the contribution of our paper in Section 3 and regarding the difference in adversarial robustness between Protonets and MAML in Section 4.1. We hope to clarify some of the reviewer’s concerns below:
>
> 1. *The downside is that this work introduces attacks, which as far as I can tell, are quite simple extensions or applications of gradient based evasion attacks such as PGD. I'm not sure if this represents a large enough contribution to warrant acceptance.*
>
>     Although our attack makes use of PGD, the fact that we’re attacking meta-learners introduces additional subtleties such that our approach is far from a straightforward application of PGD. Specifically:
>     - We generate an entire adversarial support set (or subset of the support set) at once, rather than generating just a single adversarial input.
>     - The attack is constructed with regard to an entire query dataset, rather than a single query point. Moreover, the goal is not for a single query to be misclassified, but for all future queries to be misclassified.
>     - Generation of the attack involves backpropagating the gradients through a meta-learner which itself is performing gradient based learning in the case of MAML, or is a complex set neural network in the case of Protonets or CNAPs.
>
>   In this context, it is prudent to use a fairly simple attack method in the face of significant increased complexity over the standard evasion attack setting.
>
> 2. *The main contribution seems to lie in experiments showing how little data is required to make these kinds of attacks work, but what should we infer from e.g. figure 4a beyond "if an adversary controls more data the attack is stronger"?*
>
>   Figures 4(a) and 4(b) (5(a) and 5(b) in the revised version) both illustrate that the adversary is stronger if they control more data. However, these figures also show that controlling some percentage of the data is not the only factor in the attack’s strength. Specifically, the adversarial support set produces images that can collude to fool the model, making the attack stronger without necessarily controlling more of the support set. For example, when attacking one point from each class, the adversarial support set drops the model accuracy by 71.9%, whereas the swap attack only drops the model accuracy by 40.7% - even though both control the same amount of data, the swap attack cannot generate a set of points that collude and so it is a weaker attack.
>   Another interesting point is that “spreading” adversarial points across different classes is more effective than attacking the same number of images from the same class. For example, attacking one point from each class drops the model accuracy by 71.9%, but attacking five points from one class only drops model accuracy by 26.3%.
>
>   Figure 5(b) -- which shows the effect of increasing the seed query set size on attack efficacy -- does indicate that “more data is better” in the sense that if the attacker has access to additional data to incorporate into the loss function, then the attack will be stronger. However, this does not require the attacker to control a larger portion of the support set.
>
> 3. *The downside is that this work introduces attacks, which as far as I can tell, are quite simple extensions or applications of gradient based evasion attacks such as PGD. I'm not sure if this represents a large enough contribution to warrant acceptance. The main contribution seems to lie in experiments showing how little data is required to make these kinds of attacks work, but what should we infer from e.g. figure 4a beyond "if an adversary controls more data the attack is stronger"? Is this a fair characterisation of the paper's contributions?*
>
>   The paper is aimed at meta-learning practitioners who wish to deploy few-shot learning methods on real world problems. Our goal is to show that they are especially and surprisingly vulnerable to data poisoning attacks.
>   Specifically, we show that it is possible to generate strong poisoning attacks using methods from evasion attack literature and that these adversarial support set attacks are very effective at confusing the model.
>   We also show that these poisoning attacks transfer to fine-tuners i.e. transfer between very different models with different learning dynamics, in which the poisoning attack is required to fool a training algorithm, not a static model.

---

> > ### Author Response · Authors · 2020-11-18
> > **Response to Reviewer #2 (Part 2)**
> >
> > (Continued)
> >
> > 4. *I found it curious that the UAP performed so poorly in comparison to the support attack, since the formulations are quite similar. I didn't understand how the UAP was constructed, it seems from Section 4.1 the perturbation was constructed from a separate classification task on 712 training labels from the full ImageNet dataset. Why was the UAP not constructed on the miniImageNet dataset? Or at least on a smaller task on the full ImageNet dataset? It is hard to reason about the veracity of comparison between the two attacks currently.*
> >
> >   Thank you for pointing this out! We initially decided on using UAP because it provided us with a conveniently task-independent way to generate adversarial attacks. However, as the reviewer points out, the decision-boundaries of a 712-way classification problem are different - far more brittle - than those of a 5-way problem, so the attack was largely ineffective on the 5-way problems. Unfortunately, it is unclear how to make the comparison fair without generating 500 different UAPs for each of the independent 5-way tasks. As such, we have decided to remove UAP as a baseline since it would be an unfair comparison.
> >
> >
> > 5. *Why was there no comparison with related work? For example, to (Goldblum et al., 2019; Yin et al., 2018) on the evasion attack side?*
> >
> >   The related works consider query attacks, which are not the focus of our paper. The main focus of the paper is to introduce adversarial support set attacks, which are not directly comparable to query attacks. However, we do perform some query attacks which we use to calculate the “swap attack” poisoning baseline. So we have added a comparison to the relevant results from Goldblum et al. (2019) in A.2 of the appendix. As may be expected, there was no significant difference between our query attacks and those of Goldblum et al. (2019) when using the same attack settings.
> >
> >
> > 6. *Why is ProtoNets more robust than MAML?*
> >
> >   Thanks for this question. In general, ProtoNets is a more capable classifier when compared to MAML [1] and as a result it tends to be more robust to adversarial support set attacks than MAML (as shown in Fig 5(b)). In addition, MAML uses gradient steps in order to adapt to a new task which can be sensitive to noise whereas Protonets computes the mean of the feature vectors for each class in the support set and this averaging tends to reduce noise.
> >   [1] Triantafillou, Eleni, et al. "Meta-dataset: A dataset of datasets for learning to learn from few examples." arXiv preprint arXiv:1903.03096 (2019)

---

### Official Review · AnonReviewer3 · 2020-10-29
**Simple method but missing many key-points**

**Rating:** 4
**Confidence:** 4

**Review:**

Summary:

This paper introduces Adversarial Support Set Attack, an attack that perturbs the support set samples under the few-shot paradigm. It makes use of a seed query set to find adversarial samples that maximize the loss on this set, with the hope that it will generalize to any query sample. Empirical results show that this attack is effective against a variety of few-shot algorithms and datasets. Additionally, adversarial samples constructed on traditional few-shot algorithms are empirically shown to transfer to the fine-tuning few-shot algorithm (when using similar feature extractors).

Pros:
1. This is the first paper that looks at support set poisoning as an adversarial attack in few-shot learning.

Cons:
1. Swap attack is not explained comprehensively. Is the support set used as the seed query set in this case? Since it is one of the baselines, a better explanation for the method should be included.
2. Figure 3 suggests that the seed query set is 20 times larger than the support set. This is unrealistic. Few-shot learning deals with scarce amounts of data. Under this paradigm, there are only a handful of samples per class. Assuming that an attacker has an order of magnitude more data is unrealistic.
3. The paper mentions that the UAP perturbations were calculated on a 712-way classification problem, where smaller perturbations would be more effective than on a 5-way classification problem. If UAP is a baseline, the comparison should be made fair.
4. As pointed out by the author's the transfer of adversarial samples in supervised learning is not new. It is not surprising that it happens in the case of few-shot learning as well, so that should not be counted as a major contribution.

Clarifications:
1. Does the seed query set have an overlap with the query sets sampled for the Support General evaluation? If they are randomly sampled, there is a chance that they have some samples in common, especially when the size of the seed query set is large. If there is an overlap in samples, experiments should be run when the two sets are disjoint.

---

> ### Author Response · Authors · 2020-11-18
> **Response to Reviewer #3**
>
> We thank the reviewer for reading the paper and for their insightful comments. We address each point in turn below:
>
> 1. *Swap attack is not explained comprehensively. Is the support set used as the seed query set in this case? Since it is one of the baselines, a better explanation for the method should be included.*
>
>   Thank you for pointing this out. We have added the following explanation to Section 4.1.
>
>   "When perpetrating a swap attack on a given task, we use the task's support set and query set as in Figure 1 *Query* to generate a query attack. We then 'swap' the role of the sets so that the adversarial query set is presented to the model to learn from i.e. as a support set. We evaluate the swap attack on 50 independent query sets, in the same way as the Support General scenario."
>
>   Note that we could consider a scenario similar to Support Specific, where we complete the swap by using the adversarial query set as the support set, and use the original support set as the query set, but we are really more interested in the evaluation setting, where we can compare generalization.
>
> 2. *Figure 4 suggests that the seed query set is 20 times larger than the support set. This is unrealistic. Few-shot learning deals with scarce amounts of data. Under this paradigm, there are only a handful of samples per class. Assuming that an attacker has an order of magnitude more data is unrealistic.*
>
>   We agree with the reviewer that assuming the attacker has access to significantly more data than is available in the training set may be unrealistic in a scarce data setting. We have updated Figure 4 to be more realistic by removing the case where $M = 20$ and instead added results for $M=1$ and $M=10$. The $M=10$ case still beats all the baselines, whereas the $M=1$ case beats the “Swap” baseline for Protonets, but not quite for MAML. Query attacks are not able to scale in strength by leveraging additional data, so we felt it important to demonstrate this possibility for support attacks.
>
>   Importantly, for the large scale experiments in Section 4.2, we did not augment the seed query set with additional points and instead only used the default size as dictated by the MetaDataset protocol. Even in this case, in addition to only poisoning 20% of the support set, our attack was very effective.
>
> 3. *The paper mentions that the UAP perturbations were calculated on a 712-way classification problem, where smaller perturbations would be more effective than on a 5-way classification problem. If UAP is a baseline, the comparison should be made fair.*
>
>   Thank you for pointing this out! We initially decided on using UAP because it provided us with a conveniently task-independent way to generate adversarial attacks. However, as the reviewer points out, the decision-boundaries of a 712-way classification problem are far more brittle than those of a 5-way problem, so the attack was largely ineffective on the 5-way problems. Unfortunately, it is unclear how to make the comparison more fair without generating 500 different UAPs for each of the independent 5-way tasks. As such, we have decided to remove UAP as a baseline since it would be an unfair comparison.
>
> 4. *As pointed out by the author's the transfer of adversarial samples in supervised learning is not new. It is not surprising that it happens in the case of few-shot learning as well, so that should not be counted as a major contribution.*
>
>   Although transfer of adversarial examples is a well-known phenomenon, in this paper we show transfer of a poisoning attack which has to fool a training algorithm, not a static model.  An adversarial support set must survive the fine-tuner’s gradient-based learning process in order to be effective - it was unclear to us that this would be the case. For this reason we think the transfer of our adversarial support set to the fine-tuner is a worthy contribution.
>
> 5. *Does the seed query set have an overlap with the query sets sampled for the Support General evaluation? If they are randomly sampled, there is a chance that they have some samples in common, especially when the size of the seed query set is large. If there is an overlap in samples, experiments should be run when the two sets are disjoint.*
>
>   In our experiments, the seed query set and the query sets sampled for the Support General evaluation are all mutually disjoint. We have updated the discussion in Section 4.1 to make this clear.

---

### Official Review · AnonReviewer4 · 2020-10-29
**Review for Attacking Few-Shot Classifiers with Adversarial Support Sets paper**

**Rating:** 6
**Confidence:** 3

**Review:**

The paper proposes a novel poisoning attack which is tailored for few shot learning classifiers. It performs an experimental evaluation of proposed attacks on various state of the art few shot learning / meta learning classifiers.

Overall it’s a good paper which adapts poisoning attack on a few shot learning task. Thus I recommend to accept it.

Strong points:
* proposed poisoning attack on few shot learning classifiers
* paper is well written and easy to understand
* good experimental evaluation of the method

Weak points:
* security aspects of poisoning are not discussed in the paper. In particular paper does not clearly describe goals of adversary (it’s implied that adversary wants to make model always misclassify entire test set), does not discuss capabilities of adversary (white-box vs black box).
* The main contribution of the paper is a poisoning attack, however authors talk quite a bit about adversarial examples (evasion attack) which could be distracting from the main point of the paper given that paper does not really add any new technique specifically related to evasion attack.
* proposed taxonomy of few shot learning attacks (fig 1) is not comprehensive.
* while the proposed attack is novel in a context of few shot learning, nevertheless it’s pretty straightforward generalization of poisoning attack for fully-supervised classifiers.


Recommendations on how to improve the paper:
* Add clear discussion of goals and capabilities of the adversary. If feasible, consider adaptation of the attack for different goals (i.e. change only a subset of predictions of the classifier on test set) and capabilities of the adversary (i.e. black box). Authors may refer to https://arxiv.org/pdf/1804.00308.pdf which discuss most of the necessary terminology and which is already cited by this paper.
* Make paper more focused on poisoning attack, and remove references to adversarial examples, whenever they are not needed.
* As mentioned above, proposed taxonomy of attacks is not comprehensive. It looks more like a list of attacks authors have tried, rather than comprehensive classification of all possible attacks. Thus I would recommend to change wording and call it “considered attacks” or something similar instead of “taxonomy”.

---

> ### Author Response · Authors · 2020-11-18
> **Response to Reviewer #4**
>
> We thank the reviewer for reading the paper and for their thoughtful recommendations. We have incorporated all the reviewer’s suggestions, adding a section on the adversary’s goals and capabilities, removing unnecessary references to evasion attacks and renaming figure 1. We would like to discuss some of the reviewer’s comments in more detail below:
>
> 1. *Security aspects of poisoning are not discussed in the paper. In particular paper does not clearly describe goals of adversary (it’s implied that adversary wants to make model always misclassify entire test set), does not discuss capabilities of adversary (white-box vs black box).*
>
>   Thank you for pointing this out. We’ve added the following discussion of what our adversary’s goals/capabilities are in Section 2.3:
>
>   The threat model may be summarized in terms of the adversary's assumed goal, knowledge and capabilities.
>
>     **Goal** The adversary could aim to compromise a system's integrity, availability or confidentiality. Poisoning attacks may aim to compromise system integrity - for example, backdoor poisoning (Chen et al., 2017b), where the aim is to misclassify only a specific query point at test time. In this paper, we consider poisoning attacks that reduce system availability, where the goal is simply to maximize system failure on any query point. Evasion attacks usually aim to compromise the integrity of the system by causing failure on specific query points at test time.
>
>     **Knowledge** We assume that the adversary has full knowledge of the model's internal workings - including gradients and other internal state information. We also assume that the attacker has access to enough data to be able to form a seed query set of at least the same size as the adversarial support set.
>
>     **Capabilities** When performing adversarial support set attacks, we consider an adversary who is able to manipulate some fraction of the support set. We constrain the attack-space by requiring that the adversary's modifications must be imperceptible. We achieve this by constraining the adversarial perturbation to be within some $\epsilon$ of the original image, measured using the $\ell_\infty$ norm. As a baseline, we also considered an attacker who is able to modify support set labels instead. See Other Possible Attacks in Section 3 for more details on this attack. When performing query attacks, we consider an adversary who is able to manipulate one or more points in the query set, again constrained by a maximum perturbation size.
>
> 2. *The main contribution of the paper is a poisoning attack, however authors talk quite a bit about adversarial examples (evasion attack) which could be distracting from the main point of the paper given that paper does not really add any new technique specifically related to evasion attack.*
>
>   The reviewer is correct that our adversarial support set attack is a poisoning attack. However when considering a meta-learner, there are two opportunities for perpetrating a poisoning attack: both at meta-training time and at meta-test time. Simply referring to the attack as a poisoning attack may cause ambiguity. To reduce confusion, we have changed some wording in the *Support Attack* portion of Section 3 and removed unnecessary references to “adversarial examples” and “adversarial points” throughout the paper, particularly in Section 4. There are still some references to adversarial examples where relevant, such as in our discussion of PGD and regarding the “swap attack” baseline.
>
> 3. *Proposed taxonomy of few shot learning attacks (fig 1) is not comprehensive.*
>
>   Thanks, we have changed the wording surrounding Fig 1 to be more appropriate, as recommended, calling it a “range of considered attacks” instead.
>
> 4. *While the proposed attack is novel in a context of few shot learning, nevertheless it’s pretty straightforward generalization of poisoning attack for fully-supervised classifiers.*
>
>   Generating the attack involves backpropagating the gradients through a meta-learner which itself is performing gradient based learning in the case of MAML, or is a complex set neural network in the case of Protonets or CNAPs. So we do not believe that the contribution is straightforward. With regard to existing poisoning attacks which our approach could be related to, it would be very useful if the reviewer could direct us to them if they have a specific poisoning attack in mind. We would welcome references to such literature so that we can incorporate them into the paper.

---

### Official Review · AnonReviewer1 · 2020-11-02
**Interesting domain for attacks, but lacks novelty and rigor**

**Rating:** 6
**Confidence:** 4

**Review:**

**Pros:**
+ The paper considers the construction of adversarial examples for a new learning paradigm which has practical relevance.
+ A number of possible threat models under the few-shot learning paradigm are considered.
+ The considered attack (a simple variation on PGD) is found to be effective against a variety of models and on a benchmark datasets.

**Cons:**
- The attack methodology is not particularly novel, as it is just a simple extension of standard PGD attacks.
- The writing in the paper lacks clarity. The descriptions of meta-learning algorithms are not clear enough for a reader with knowledge of supervised learning but limited background on few-shot learning. In particular, there should be a dedicated section explaining the differences between attacks on traditional supervised learning and few-shot learning.
- The related work refers to a couple of previous papers that have explored defenses against the query-based threat model for misclassifying particular examples. Similarly, this paper should have explored techniques for defending against the proposed support-set attacks. For example, the defender may add steps of training on the poisoned task to 'undo' the effects of support set attacks
- The attack success metrics are not clearly defined. How are the 500 tasks mentioned in figure 3 chosen? Is the support set data at for the new tasks at test time modified at all? How does attack success vary with the similarity of tasks at training time vs those at test time?

---

> ### Author Response · Authors · 2020-11-18
> **Response to Reviewer #1 (Part 1)**
>
> We thank the reviewer for taking the time to read and review our paper. We appreciate the comments and discuss each point in turn below:
> 1. *The attack methodology is not particularly novel, as it is just a simple extension of standard PGD attacks.*
>
>   Although our attack makes use of PGD, the fact that we’re attacking meta-learners introduces additional subtleties such that our approach is far from a straightforward application of PGD. Specifically:
>     - We generate an entire adversarial support set (or subset of the support set) at once, rather than generating just a single adversarial input.
>     - The attack is constructed with regard to an entire query dataset, rather than a single query point. Moreover, the goal is not for a single query to be misclassified, but for all future queries to be misclassified.
>     - Generation of the attack involves backpropagating the gradients through a meta-learner which itself is performing gradient based learning in the case of MAML, or is a complex set neural network in the case of Protonets or CNAPs.
>
>   In this context, it is prudent to use a fairly simple attack method in the face of significant increased complexity over the standard evasion attack setting.
>
> 2. *The writing in the paper lacks clarity. The descriptions of meta-learning algorithms are not clear enough for a reader with knowledge of supervised learning but limited background on few-shot learning. In particular, there should be a dedicated section explaining the differences between attacks on traditional supervised learning and few-shot learning.*
>
>   The intended audience for this paper are meta-learning practitioners who wish to deploy these few-shot learning methods on real world problems. Our goal is to show that surprisingly they are especially vulnerable to data poisoning attacks. With that in mind, we only briefly reviewed the meta-learning literature. However, the reviewer is correct that a wider readership could be interested in the paper and so we have made use of the extra space we now have to extend the description of the meta-learning approaches we study in Section 2.1.
>
> 3. *The related work refers to a couple of previous papers that have explored defenses against the query-based threat model for misclassifying particular examples. Similarly, this paper should have explored techniques for defending against the proposed support-set attacks. For example, the defender may add steps of training on the poisoned task to 'undo' the effects of support set attacks*
>
>   We agree that defence mechanisms are an interesting line of research to pursue. It is possible in principle to use adversarial training to improve robustness to context set attacks, however such context set attacks are significantly more expensive than standard attacks (for the reasons outlined in our response to point 1) and so various approximations / accelerations are likely to be required to use these attacks within a training procedure.
>
>   Both other papers that we cite pursue a form of adversarial training or hardening by exposing the meta-learner to adversarial examples during meta-training. However, these attacks are simple query point attacks, whereas the support set attack developed in our paper requires the significant overhead of  backpropagating the gradients through a meta-learner to a set of support points.
>
>   Even though the attacks in these papers are less expensive than the support set attacks, the authors find it necessary to take steps that reduce the attack’s computational cost to be feasible. Yin et al. only considers FGM [1], a very simple, single step method of attack generation, which is not considered a good test for robustness [2]. Additionally, the adversarial examples are generated with respect to a fixed set of model parameters, and so does not incur the additional cost of considering the learning dynamics of the meta-learner when doing adversarial training.
>
>   Goldblum et al. largely restrict their experiments to augmenting the query sets with adversarial points because of the computational cost of augmenting the support sets at well. (They justify this by an experiment on MAML, where they find that augmenting the support sets does not significantly improve robustness against evasion attacks). However, since our attack relies on poisoning the support set, such shortcuts are likely not applicable to adversarial support set attacks.
>
>   [1] Ian J. Goodfellow, Jonathon Shlens, Christian Szegedy, (2015), “Explaining and Harnessing Adversarial Examples”, available at: https://arxiv.org/pdf/1412.6572.pdf
>
>    [2] Nicholas Carlini, et al., (2019), “On Evaluating Adversarial Robustness”, available at: https://arxiv.org/pdf/1902.06705.pdf

---

> > ### Author Response · Authors · 2020-11-18
> > **Response to Reviewer #1 (Part 2)**
> >
> > 4. *The attack success metrics are not clearly defined.*
> >
> >   As mentioned in Section 4.1, we measure the attack’s success against a model in terms of the percentage relative decrease in the model’s classification accuracy as follows:
> >     $100  \times (a_{clean} - a_{attack})/a_{clean}$ where $a_{clean}$ is the clean classification accuracy, and $a_{attack}$ is the classification accuracy after the attack.
> >
> >   Using a relative measure allows us to compare the effect of our attack across models and across tasks with different shot and way. We consider our attack successful if it performs better than the baseline attacks in terms of the percentage relative decrease in model accuracy, as defined above.
> >
> > 5. *How are the 500 tasks mentioned in figure 4 chosen?*
> >
> >   Each task is constructed randomly for the given shot and way, i.e. by choosing the required number of classes from the dataset labels at random, then choosing the required number of images randomly within each class. Each task is composed of a support set, a seed query set and 50 unseen query sets used for evaluation of the attack. Although the shot of the seed query set may be varied between experiments, the query sets used for evaluation are the same size as the support set. The unseen query sets are all guaranteed to be disjoint from the seed query set to avoid information leakage.
> >
> >   We have added these details to Section 4.1.
> >
> > 6. *Is the support set data at for the new tasks at test time modified at all? How does attack success vary with the similarity of tasks at training time vs those at test time?*
> >
> >   We generate an adversarial support set using the corresponding seed query set for a given task. We then evaluate the adversarial support set on the task’s 50 unseen query sets. We make no further changes to the adversarial support set when evaluating it on the unseen query sets.
> >
> >   We don’t consider the effect of transferring an adversarial support set from one task to a completely different task. Since the tasks are generated randomly, the composition of the tasks -- in terms of the classes and images within those classes -- are entirely different from one another and it is not clear what to expect.
> >
> >   We have updated Section 4.1 to provide a clearer explanation of the experimental methodology regarding different tasks. We hope this answers the reviewer’s questions regarding the relationship between different tasks.

---

### Author Response · Authors · 2020-11-18
**General Response**

We thank all the reviewers for their comments which we have used to improve the paper. Two of the reviewers are in favor of accepting the paper, but two of the reviewers have concerns. All reviewers had a common concern about the magnitude and novelty of the technical contribution as the paper builds on what is a fairly basic attack strategy in the normal adversarial example setting (projected gradient descent, PGD).

We believe that this characterization overlooks the significant added complexity of the setting we consider. Specifically,
- We generate an entire adversarial support set (or subset of the support set) at once, rather than generating just a single adversarial input.
- The attack is constructed with regard to an entire query dataset, rather than a single query point. Moreover, the goal is not for a single specific query to be misclassified, but for all future unseen queries to be misclassified.
- Generation of the attack involves backpropagating the gradients through a meta-learner which itself is performing gradient based learning in the case of MAML, or is a complex set neural network in the case of protonets or CNAPs.

In this context, it is prudent to use a fairly simple attack method in the face of significant increased complexity over the standard evasion attack setting.

Here we quickly summarize the changes we have made to the paper in response to the reviewer comments:
- We have added a paragraph to Section 3, emphasizing the technical contribution of our proposed attack and how it differs from usual application of PGD.
- As pointed out by two of the reviewers, UAP is not a realistic comparison and has been removed as a baseline from all results.
- Figure 4, illustrating the relative drop in model accuracy in the small-scale scenario, has been updated to include the results for the support set attack in the case where $M=1$ and $M=10$. The case where $M=20$ has been removed, so that we now focus on scenarios where the attacker has access to less data. We also updated the discussion of Figure 4 to consider the differences between $M=1$ and $M=10$.
- We have revised the paper’s nomenclature, specifically regarding the terms “adversarial examples” and “adversarial patterns” to be less ambiguous, especially throughout Section 4.
- We added a discussion of the attack scenario i.e. the adversary’s goals, knowledge and capabilities in Section 2.3.
- We provided additional experimental details regarding construction of small-scale tasks in Section 4.1. We also clarified the relationship between the support set, seed query set and evaluation query sets.
- We provided more experimental details about how “Swap attacks” are implemented in Section 4.1.
- We have added further explanation of the meta-learning algorithms used in Section 2.1, under “Common Few-shot Learning Algorithms”.
- We added a short discussion on the difference between Protonets and MAML in terms of robustness to Section 4.1, under “Seed Query Set Size”.
- We have changed the wording around Figure 1 to be more precise.
- We have extended A.2 of the appendix to include a comparison between our PGD query attack and the results from Goldblum et al. (2019).

---

### Decision · Program_Chairs · 2021-01-07
**Final Decision**

**Decision:**

Reject

**Comment:**

This paper presents a method for attacking few-shot learners with poisoning a subset of support set. I believe this might be the first work to address adversarial examples for meta-learners (or few-shot learners), which is a timely issue. A common concern raised by most of reviewers is in the novelty of this work, in the sense that the method builds on a basic attack strategy (such as PGD) in the standard adversarial example setting. Authors responded to this, summarizing what's new in this paper. Episodic training for few-shot learners requires consuming support set (instead of single training data point). It is a nature of most meta-learning methods. Thus, it is easily expected that the adversarial attack for few-shot learners is naturally extended to poisoning a support set (or its subset) instead of a single data point. Certainly such extension may entail a new strategy. However, during the discussion period with reviewers, concerns on the novelty of such extension still remains. In particular, the few-shot learning algorithms do not allow big changes in the original model. The algorithms analyzed are prototypical networks that do not utilize fine-tuning, and MAML that fine-tunes for a small number of pre-fixed steps. So the transfer of adversarial samples may not be counted as a major contribution.